# Recent Trends in the Application of Oilseed-Derived Protein Hydrolysates as Functional Foods

**DOI:** 10.3390/foods12203861

**Published:** 2023-10-21

**Authors:** Katarzyna Garbacz, Jacek Wawrzykowski, Michał Czelej, Tomasz Czernecki, Adam Waśko

**Affiliations:** 1Biolive Innovation Sp. z o. o., 3 Dobrzańskiego Street, 20-262 Lublin, Poland; 2Department of Biotechnology, Microbiology and Human Nutrition, Faculty of Food Science and Biotechnology, University of Life Sciences in Lublin, Skromna 8, 20-704 Lublin, Poland; 3Department of Biochemistry, Faculty of Veterinary Medicine, University of Life Sciences in Lublin, Akademicka 12, 20-033 Lublin, Poland

**Keywords:** oilseed, bioactive peptide, functional food, protein hydrolysate

## Abstract

Oilseed-derived proteins have emerged as an excellent alternative to animal sources for the production of bioactive peptides. The bioactivities exhibited by peptides derived from plant proteins encompass a wide range of health-promoting and disease-preventing effects. Peptides demonstrate potential capabilities in managing diseases associated with free radicals and regulating blood pressure. They can also exhibit properties that lower blood sugar levels and modify immune responses. In addition to their bioactivities, plant-derived bioactive peptides also possess various functional properties that contribute to their versatility. An illustration of this potential can be the ability of peptides to significantly improve food preservation and reduce lipid content. Consequently, plant-derived bioactive peptides hold great promise as ingredients to develop functional products. This comprehensive review aims to provide an overview of the research progress made in the elucidation of the biological activities and functional properties of oilseed-derived proteins. The ultimate objective is to enhance the understanding of plant-derived bioactive peptides and provide valuable insights for further research and use in the food and medicine industries.

## 1. Introduction

The global food market is undergoing a dynamic transformation as a result of accelerating climate change, the depletion of natural resources, and the growing population. Currently, the main trends observed in the food industry place emphasis on products known as superfoods, sustainable food production, and plant-based foods. Consumers are increasingly looking for products that are safe and natural and are produced using environmentally friendly technologies [1]. This has contributed to the growing interest of the social and scientific community in functional foods, defined as processed or natural foods that, when consumed regularly, show documented positive effects on the human body in addition to the nutritional effects of the nutrients contained therein that are considered essential. The action of functional foods is to improve health and reduce the risk of certain diseases, i.e., cancer, type 2 diabetes, stroke, and cardiovascular disease (CVD) [2]. To achieve this goal, bioactive compounds with, e.g., antimicrobial, antioxidant, or immunomodulatory effects which would constitute functional foods, in combination with conventional food products, are being sought [1]. Research carried out in recent years indicates that protein hydrolysates have great potential in this aspect [3]. 

Protein hydrolysates are defined as a complex mixture of oligopeptides, peptides, and free amino acids which are formed by partial or complete hydrolysis. In turn, bioactive peptides (BPs) are characterized as a group of organic compounds possessing numerous health-promoting properties such as antioxidant [4], immunomodulatory, antihypertensive, antimicrobial [5], and hypoglycaemic [6] effects as well as cholesterol-lowering [7] or anti-cancer activity [8]. The bioactive potency of these compounds depends not only on the composition and sequence of amino acid residues but also on their hydrophobicity and charge [3,9]. These compounds can be obtained by digestion in the gastrointestinal tract, fermentation, enzymatic hydrolysis, or in vitro chemical reactions [10]. Protein hydrolysates are obtained from a variety of animal, fungal, or plant sources [11,12,13,14]. Given the growing focus on plant-based foods, the use of protein hydrolysates extracted from oilseeds has received considerable interest [15]. 

Oilseed crops are used in the food industry, e.g., for the production of margarine, confectionery products, or unrefined edible oils [16]. The most commonly cultivated oilseed crops include rapeseed/canola (*Brassica napus* L.), sunflower (*Helianthus annuus* L.), soy (*Glycine max* L.), olive (*Olea europaea* L.), and peanut (*Arachis hypogaea* L.). Their seeds are rich in unsaturated fatty acids, sterols, fiber, proteins, and biopeptides, making them excellent functional ingredients [15]. The effects of oils on human health are invaluable. They not only ensure the absorption of fat-soluble vitamins but also support the efficient operation of the endocrine and neurotransmission systems [16]. The production of cold-pressed oil yields a fiber- and protein-rich by-product called cake, which is increasingly being used as an ingredient in the production of fortified foods. The search for new sources of plant-derived protein hydrolysates for use in food is crucial given the ethically and environmentally justified need to reduce the consumption of meat products, which are the primary source of protein in conventional diets [17].

Hence, in the present work, we focus on different protein hydrolysates with documented health-promoting properties derived from different types of oilseed proteins and their potential use in functional foods. Our aim was to compile recent developments in this field and to identify key steps in the production of foods enriched with protein hydrolysates derived from oilseed plants.

## 2. Biological Activity of Oilseed-Derived Bioactive Peptides

Dietary proteins, including oilseed proteins, are a source of not only valuable amino acids but also bioactive peptides. Biologically active peptides are fragments of the amino acid sequence of food proteins that become active when released. They are usually released during digestion, fermentation (due to the proteolytic activity of microorganisms), or in vitro enzymatic processes and can then have an impact on human health (Figure 1). Various factors such as the protein type and source, protein pre-treatment, enzyme type, and proteolysis conditions can alter the functionality of hydrolysates and bioactive peptides [18]. Table 1 shows a list of findings of sources of oilseed-derived peptides along with their bioactive properties.

### 2.1. Antioxidant Activity

Free radicals are one of the etiological factors of many so-called ‘diseases of civilisation’, including cardiovascular disease, diabetes, cancer, diabetes, or rheumatic diseases. For this reason, synthetic antioxidants are used to protect the damaging effects of free radicals in the body [27]. However, studies have shown that synthetic antioxidants can be toxic and dangerous for human health [21]. Therefore, natural antioxidants exhibiting antioxidant potential with little to no side effects have been extracted from various materials. Numerous scientific studies have highlighted the antioxidant properties of protein hydrolysates derived from diverse sources [10]. Free radicals or reactive oxygen species (ROS) can interact with amino acids. However, aromatic amino acids (tyrosine, phenylalanine, and tryptophan), imidazole-containing amino acids (histidine), and nucleophilic sulfur-containing amino acids (methionine and cysteine) have demonstrated the highest reactivity [42]. In a study conducted by Zhang et al. [32], it was shown that a soy protein hydrolysate had the ability to scavenge the free radicals DPPH (2,2-diphenyl-1-picryl-hydrazyl-hydrate) (IC50 = 4.22 mg/mL) and ABTS (2,2′-azinobis-(3-ethylbenzothiazoline-6-sulfonate)) (IC50 = 2.93). Furthermore, the study showed that the soy protein hydrolysate inhibited the production of intracellular reactive oxygen species (ROS) in Caco-2 cells. Also, Yu et al. [33] investigated the antioxidant activity of four soybean meal peptide fractions (PF1: >5 kDa; PF2: 3–5 kDa; PF3: 1–3 kDa; PF4: <1 kDa). The authors demonstrated that the 1–3 kDa peptides exhibited the highest antioxidant activity. In their study, Lu et al. [31] identified seven novel antioxidant peptides derived from sesame protein hydrolysates, with SYPTECRMR, whose IC50 values for DPPH and ABTS were 0.105 mg/mL and 0.004 mg/mL, respectively, showing the highest antioxidant activity. Furthermore, the authors concluded that the presence of Cys6, Met8, the bulky C-terminal amino acid residue (Arg9), and the negatively charged group around sulfur-containing amino acids were responsible for the antioxidant activity of SYPTECRMR. The rapeseed peptides obtained by fermentation showed high free radical scavenging activity, reducing dpower and inhibition of lipid peroxidation, but present low iron ion chelating activity [27]. In a study conducted by Kamarać et al. [21], the antioxidant activity of flaxseed protein hydrolysates derived from five different enzymes was evaluated. The results showed that there was a slight variation in antioxidant activity among the hydrolysates, with the FRAP (ferric reducing antioxidant power) values ranging from 0.20 to 0.24 mmol Fe^2+^/g and the ABTS scavenging activity ranging from 0.17 to 0.22 mmol Trolox/g. In research undertaken by Girgih et al. [25], they subjected a hemp protein isolate to hydrolysis by using the enzymes pepsin and pancreatin, resulting in the generation of bioactive peptides. The authors proceeded to identify the sequences for 23 peptides within the hydrolysate. Further examination through in vitro and in vivo testing revealed the enhanced antioxidant characteristics of two peptides, WVYY and PSLPA.

### 2.2. ACE Inhibitor Activity

Inhibition of the angiotensin I-converting enzyme (ACE; EC 3.4.15.1) by certain food peptides contributes to lowering blood pressure in hypertensive patients. Peptides with ACE-inhibiting activity used as dietary components may support hypertension therapy [43]. As indicated in the literature, the inhibitory activity of peptides against ACE is closely related to their structure. The chain length, composition, and amino acid sequences are crucial [44]. ACE-inhibitory peptides are generally short-chain peptides consisting of specific amino acid residues at the N and/or C terminus. The presence of aromatic (phenylalanine, tryptophan or tyrosine, proline), hydrophobic (leucine, isoleucine, and valine), and basic (arginine and lysine) amino acids at the C terminus has been found to have a strong effect on ACE binding [45]. Available research highlights the possibility of using protein hydrolysates derived from oilseeds as ACE inhibitors. In a study conducted by Puchalska et al. [34], the soy peptide VLIVP isolated from soy protein hydrolysates by protease P showed higher ACE inhibitory activity than the well-studied IPP and VPP peptides from milk. Tsai et al. [35] conducted a study into the potential mechanism of a bioactive peptide from soybean, (VHVV), through an in silico model and spontaneously hypertensive rat experiments. Their docking study revealed that the VHVV peptide from soybean possesses the ability to connect with the ACE active site, thereby inhibiting its activation. Segura-Campos et al. [20], who obtained bioactive peptides from chia proteins by controlled protein hydrolysis using the Alcalase-Flavourzyme sequencing system, obtained results indicating that amino acid hydrophobic residues contributed significantly to the ACE-I inhibitory potency of the chia peptide, probably by blocking angiotensin II production. The inhibitory activity ranged from 48.41% to 62.58% in the purified fractions, but the fraction with a molecular weight of 1.5–2.5 kDa showed the greatest inhibition potential (IC50 = 3.97 μg/mL; elution volume 427–455 mL). Research conducted by Marambe et al. [22] revealed that potent angiotensin-converting enzyme inhibitory activities were exhibited by flaxseed proteins when hydrolyzed with Flavourzyme, as indicated by IC50 values of 0.07 mg/mL. Subsequent investigations by Nwachukwu et al. [23] signal that this flaxseed protein hydrolysate displayed marked inhibition on the function of the angiotensin I-converting enzyme, while concurrently demonstrating a minimal suppressive impact on renin secretion in hypertensive rat models. Orio et al. [26] subjected a hemp seed protein isolate to intensive chemical hydrolysis, and then the purified fractions were tested as angiotensin-converting enzyme inhibitors. The authors identified four potentially bioactive peptides, GVLY, IEE, LGV, and RVR. The IC50 values for these peptides were determined in the range of 16–526 μM, confirming that hemp seed may be a valuable source of hypotensive peptides. A study conducted by Duan et al. [28] aimed to evaluate bioactive peptides in rapeseed protein and identify novel angiotensin I-converting enzyme inhibitory peptides using bioinformatics methods. The authors identified the novel peptides FQW, FRW, and CPF, which exerted potent inhibitory effects on ACE in vitro with IC50 values of 44.84 ± 1.48 μM, 46.30 ± 1.39 μM, and 131.35 ± 3.87 μM, respectively. He et al. [29] discovered that the ACE inhibitory peptides GHS, RALP, and LY, derived from canola protein, exhibited considerable inhibitory effects on nitric oxide secretion and proinflammatory cytokines. Additionally, these peptides ameliorated cell damage instigated by oxidative stress in spontaneously hypertensive rats.

### 2.3. Antimicrobial Activity

Given the rising resistance of microbes to synthetic antibiotics, there is a need to explore new bioactives or extracts as natural and safer alternative antimicrobial agents to treat microbial infections [46]. Antimicrobial peptides consist of up to 50 amino acid residues. In natural conditions, these usually cationic peptides contain hydrophobic amino acids and exhibit a broad spectrum of activity against bacteria, viruses, and fungi [47]. Despite intensive research, their mechanism of action is not fully understood. The most important effects of these peptides include changes in cell membrane permeability, destabilization of membrane lipid structure, binding to lipopolysaccharide, inhibition of DNA replication, inhibition of protein expression, and release of ATP, thus leading to cell lysis [48]. Bioactive peptides derived from oilseeds were also tested for antimicrobial activity [46]. Volatile oils from different *Nigella sativa* samples were examined for their effectiveness against several bacterial isolates. A clear zone of inhibition of growth was observed in the case of *Staphylococcus aureus*, the development of which was associated with the two important bioactive ingredients of *N. sativa*—thymoquinone and melanin [49]. In another study, the antifungal activity of methanolic, aqueous, and chloroform extracts of *N. sativa* was compared. The methanolic extracts were found to have the strongest effect against several strains of *Candida albicans*, followed by the chloroform extracts, while the aqueous extracts did not exhibit antifungal activity [50]. Also, certain proteins and peptides isolated directly from Cucurbitaceae seeds showed antifungal and antimicrobial properties [51]. For example, a peptide (PQRGEGGRAGNLLREEQEI) with a molecular weight of 8 kDa isolated from black pumpkin seeds inhibited mycelial growth in the fungi *Botrytis cinerea*, *Fusarium oxysporum*, and *Mycosphaerella oxysporum* [19]. Freitas et al. [36] isolated twelve bioactive peptides from soybean meal protein that inhibited the growth of food-borne pathogenic bacteria. In addition, the authors showed that the isolated peptides were not toxic to mouse fibroblast and bone marrow cells. Liu et al. [52] reported antimicrobial activity of cyclinopeptides present in flaxseed oil. The authors demonstrated the antibacterial activity of 1-Mso-cyclolinopeptides B and 1-Mso, 3-Mso-cyclolinopeptides F against *Listeria monocytogenes*. The research of Dhayakaran et al. [37] provided evidence that the soy peptides PGTAVFK and IKAFKEATKVDKVVVLWTA exhibit antimicrobial properties against *P. aeruginosa* and *L. monocytogenes*.

### 2.4. Hypolipidaemic Activity

Hyperlipidemia is a prevalent metabolic disorder characterized by elevated levels of lipids in the bloodstream; it is associated with an increased risk of cardiovascular diseases such as myocardial infarction and atherosclerosis [30]. Available scientific research indicates that bioactive peptides contained in oilseeds have the potential to lower blood cholesterol levels [38,39]. Using gel filtration chromatography–mass spectrometry, Yang et al. [30] identified rapeseed peptides with the amino acid sequence EFLELL which exhibited good hypolipidaemic activity. The study evaluated the IC50 values of the EFLELL peptides, which were 0.1973 ± 0.05 mM (sodium taurocholate), 0.375 ± 0.03 mM (sodium cholate), and 0.203 ± 0.06 mM (glycine sodium cholate). The hypolipidaemic activity of the EFLELL peptides was further investigated using cell lines, and the results indicated a significant decrease in total cholesterol (T-CHO), triglycerides (TG), and low-density lipoprotein cholesterol (LDL-C) under the influence of the rapeseed peptides. Aiello et al. [38] determined the presence of three peptides, LPYP, IAVPTGVA, and IAVPGEVA, derived from soy protein, which demonstrated hypocholesterolemic capabilities. These peptides have the ability to inhibit the functions of a key enzyme (3-hydroxy-3-methylglutaryl coenzyme A reductase, HMG-CoA reductase) implicated in the biosynthesis of cholesterol and, furthermore, modulate cholesterol metabolism in HepG2 cells. Similarly, Lammi et al. [39] discovered two other hypocholesterolemic peptides—YVVNPDNDEN and YVVNPDNNEN—also produced from soy βCG. These peptides additionally regulate cholesterol through the same method.

### 2.5. Immunomodulatory Activities

While the immune-modulating activities of dairy protein peptides are well known, similar peptides derived from plant proteins through enzymatic hydrolysis have only recently become more available [41]. Velliquette et al. [41] investigated the anti-inflammatory and immunomodulatory properties of a sunflower protein hydrolysate. The authors identified four novel peptides, YFVP, SGRDP, MVWGP, and TGSYTEGWS, which inhibited IL-1β-mediated NF-κB activation. The MVWGP peptide showed the strongest immunomodulatory effect, which may be related to the presence of methionine residues. Wen et al. [40] used Alcalase^®^ and neutrase to produce bioactive peptides from soy protein isolates. The researchers identified eighty-five peptide sequences, eighty-four of which could be involved in immunomodulatory properties. These specific peptides were found to have a role in adjusting the activities of cytokines such as TNF-α and IL-6. Additionally, they could also facilitate the propagation of macrophages and augment the concentration of nitric oxide, an immune response mediator. Udenigwe et al. [24] used the enzymes pepsin, ficin, and papain to digest flaxseed proteins. The fraction of peptides with a molecular weight below 1 kDa was separated. These peptides demonstrated notable suppression effects on the nitric oxide production instigated by lipopolysaccharides in RAW 264.7 macrophages, with no observable cytotoxicity.

## 3. Use of Oilseed-Derived Protein Hydrolysates as Functional Foods

In recent years, there has been a significant increase in consumer interest in food products with functional properties [53]. Due to the extensive effects of bioactive peptides derived from oilseeds, they are more often analyzed for use in functional foods. In addition, the use of cakes in food, which are a by-product of oil production, as an alternative source of protein may ensure the management of waste from the food industry [15]. Moreover, products enriched with proteins of plant origin are good alternatives for vegans and people with dairy allergies [54]. Protein hydrolysates can be used in the production of protein-rich foods, i.e., protein drinks, protein-rich pasta, powdered drinks, infant and weaning foods, bars, or meat substitutes (Figure 2) [55,56,57,58,59]. 

Oilseed proteins and peptides are generating considerable interest in the functional food industry due to their robust bioactive properties, including antioxidant, antihypertensive, and neuroprotective activities [15]. They also provide a balanced profile of amino acids that are ideal in diverse sectors, ranging from baking to the meat industry [60,61]. A notable advantage of many oilseeds is their low allergenicity or even non-allergenic properties, making them valuable in the development of functional food products [15]. Particular emphasis must be placed on the profound usage potential of bioactive peptides and protein hydrolysates as natural food preservatives, given their antimicrobial and antioxidant properties [14]. The research conducted by Ospina-Quiroga [55] underscored the pivotal function of hydrolysates derived from oilseed plants as potent antioxidants in food emulsions. Hydrolysates derived from sunflower, rapeseed, and lupin were recognized as efficacious emulsifying agents. These significantly inhibited lipid oxidation, which is a primary factor influencing food quality and longevity [55]. Further, earlier research by Zhang et al. [56] emphasized the role of soy protein hydrolysates in reducing lipid peroxidation. It was observed that the incorporation of bioactive peptides derived from three different fractions of soy protein hydrolysates, which were prepared using microbial proteases, resulted in a noteworthy decrease in lipid peroxidation in ground beef samples. The study conducted by Lee et al. [61] underscores the role of soy hydrolysates as potential antioxidants in the production of specific food products such as pork patties. Research conducted by Hou et al. [57] identified that glycinin basic polypeptides derived from soybean exhibit potent antifungal properties. These polypeptides demonstrated the capability to effectively hinder mycelial growth and spore germination, achieved through the disruption of fungal plasma membranes via the ergosterol synthesis interference. This suggests their potential use in food preservation, improving the sensory qualities of wet noodles. In a similar vein, another study from Ning et al. [62] discovered the role that soy peptides play in enhancing the texture of Scomberomorus niphonius surimi (Japanese Spanish Mackerel) and significantly diminishing microbial growth to extend the product’s shelf life. These findings, thus, underline the potential that soybean-derived peptides offer as bioactive food additives, particularly in the context of starchy food items and surimi products. In their research, Segura-Campos et al. [63] found that incorporating chia protein hydrolysates into items like white bread and carrot cream was associated with a marked enhancement in ACE inhibitory activity, compared to the unmodified food. Such findings indicate the implications of chia protein hydrolysates in the development of functional food, mainly due to its apparent utility in controlling high blood pressure. Recent investigations have also highlighted the promising applications of hydrolysates, derived from oilseed plants, in the functional beverage industry. Sarker [53] described the implementation of sesame peptides in the development of beverages designed to serve as anti-hypertensive agents. According to Fan [54], there have been various efforts to create functional drinks containing soybean peptides in order to enhance their stability. As a case in point, a novel health drink was developed by Zhang et al. [64] by infusing soybean peptide and selenium-rich yeast into the beverage composition. In the study conducted by Puchalska et al. [58], it was indicated that infant formulas based on soybean peptides were interesting alternatives to cow’s milk infant formula. In addition, these peptides were also reported to possess high stability in thermal and acid treatment, good emulsifying properties, and low viscosity. 

Beyond the bioactive properties of hydrolysates, hydrolysis also instigates structural adjustments in proteins which improve their solubility, surface active properties, hydration and gelling potential, and overall functionality [17]. The research conducted by Chen et al. [65] illustrated that ice cream’s interfacial and viscoelastic properties can be notably improved via the utilization of soy protein hydrolysates obtained using papain and pepsin enzymes. This enhancement results in superior emulsion stability and decelerates the melting rate, thus outperforming the use of skimmed milk powder in ice cream production. Another study concerned with using hydrolysates in ice cream was conducted by the Liu et al. [66]. It was found that soy protein hydrolysates, combined with xanthan gum, were effective as substitutes for fat in low-fat ice cream. The investigators resorted to a combined methodology of enzymatic hydrolysis and thermal-shearing treatment to procure the desired properties. The potential of canola protein hydrolysates in the sector of meat products has been highlighted by Karami and Akbariadergani [67]. Their study showed that canola protein hydrolysates improve the cooking yield, owing to their water-holding capacity. Moreover, Aluko and McIntosh [68] demonstrated that canola protein hydrolysates, having a hydrolysis degree (DH) between 7% and 14%, managed to effectively substitute between 20% and 50% of the egg content in various mayonnaise recipes. Additionally, research by Guo et al. [69] shed light on the implicative usage of enzymatic hydrolysates extracted from canola proteins in the production of meat-flavored seasonings. They suggested that the creation of ingredients that mimic the aroma of cooked meat could be achieved by conducting the process at lower temperatures and pH values. Conversely, the scent of roasted meat could be achieved by implementing higher temperatures. This showcases the versatility of bioactive hydrolysates from oilseed plants in modifying food flavor profiles.

Hydrolysates of proteins can be incorporated into food products to enhance their nutritional profile. In a work conducted by Guo et al. [70], soy protein hydrolysates were used to fortify wheat flour, resulting in a marked reduction in gluten content and an improvement in the nutritional value of noodle dough. This, in turn, magnified the quality and nutritional attributes of the noodles. Similarly, the research executed by Schmiele et al. [60] exhibited the application of soy protein hydrolysates in bread and bakery products. This additive led to increased firmness while concurrently improving the nutrition level of these bakery products. Pap et al. [71] conducted a study investigating the enzymatic hydrolysis of hemp seed cake, which resulted in two fractions: sediment and liquid. The sediment, possessing the bulk of the major components, exhibited a promising potential for use in solid food formulations, such as breads, crispbread, or crackers. Conversely, the liquid facet, boasting high solubility, showcased its appropriateness for assimilation into drinks and liquid foods, thereby amplifying their nutritional worth.

Lots of examples exist where processed oilseed cake has been used in food. Łopusiewicz et al. [59] used flaxseed cake to produce new fermented beverages such as kefir. The authors obtained results confirming that lactic acid bacteria and yeast were able to grow well in flaxseed cake without any supplementation, and their viability exceeded the recommended level for kefir products. Therefore, beverages can be used as a new non-dairy agent to support beneficial microflora. Research was also conducted on the use of oil cake in the production of bakery products such as bread [72] or biscuits [73]. In order to produce enriched bread, the authors replaced part of the wheat flour with oil and walnut cake (1%, 3%, 5%). Bread containing the highest addition of cake was characterized by the highest hardness as well as the highest antioxidant activity. 

Plant-derived proteins applied in the food industry can also be used in the production of food packaging materials [74]. Tkaczewska [75] indicated that some protein hydrolysates that have antimicrobial and antioxidant activities can be used as natural food preservatives. In addition, these peptides can be used as active ingredients in packaging materials such as coatings and edible films. Suput et al. [76], who investigated the possibility of using sunflower oil cake for the production of biopolymer films, obtained smooth and flexible dark brown-green films. Another study demonstrated the potential of rapeseed peptides to be used as a carrier for β-carotene encapsulation [77].

## 4. Summary

The increased consumer awareness of a healthy diet has led to the rapid development of products known as functional foods in the global food market [1]. New solutions are being sought to provide suitable nutrients to the diet and to benefit human health. An additional challenge is to satisfy the expectations of consumer groups with specific needs, such as vegans, vegetarians, or people with food intolerances. Oilseeds are a valuable source of not only nutrients but also bioactive peptides, offering potential health benefits. These peptides show promising applications in mitigating diseases related to free radicals and in controlling blood pressure. Furthermore, their potentials in food preservation and the reduction in lipid levels have also been noted. Studies have revealed the antioxidant properties of protein hydrolysates from numerous oilseed sources. Their importance in maintaining cardiovascular health, through inhibiting the angiotensin I-converting enzyme (ACE), has also been highlighted. Moreover, peptides derived from oilseeds have shown an antimicrobial effect and also hypolipidemic ability. Due to their numerous properties, oilseed-derived peptides and protein hydrolysates have been incorporated in various food products like functional beverages, bakery products, and meat and dairy substitutes, enhancing their nutritional profile and quality. They also have potential applications in the food packaging industry. This review highlights the importance of research into bioactive peptides derived from oilseeds as they offer potential health benefits and possible use in the food industry.

## Figures and Tables

**Figure 1 foods-12-03861-f001:**
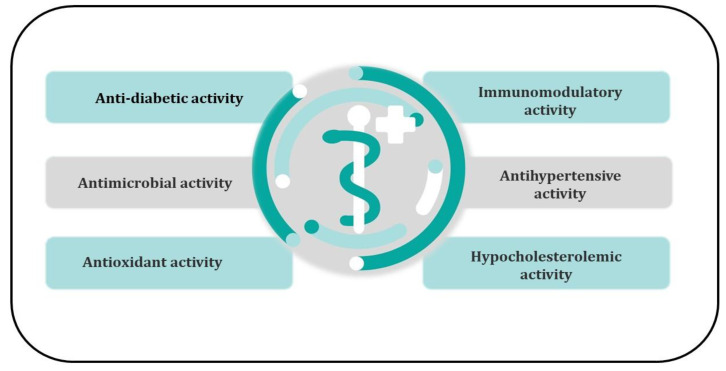
Beneficial effects of bioactive peptides.

**Figure 2 foods-12-03861-f002:**
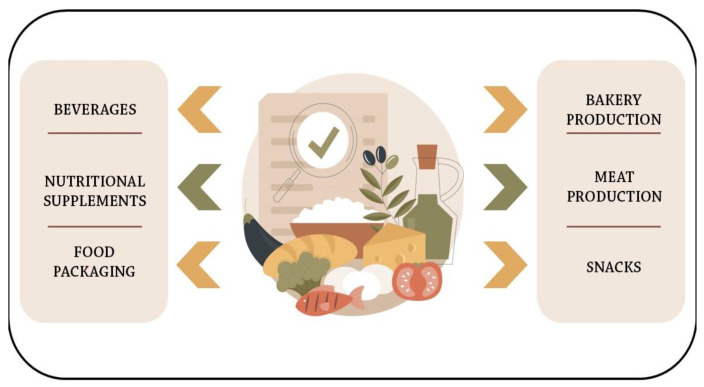
Potential use of oilseed protein hydrolysates in functional food.

**Table 1 foods-12-03861-t001:** Peptides from oilseed plants and their bioactivity.

Source	Peptide Sequence	Bioactive Properties	References
Black pumpkin	PQRGEGGRAGNLLREEQEI	antimicrobial	[19]
Chia	-	ACE inhibitor	[20]
Flaxseed	-	antioxidant	[21]
Flaxseed	-	ACE inhibitor	[22,23]
Flaxseed	-	immunomodulatory	[24]
Hemp	WVYY, PSLPA	antioxidant	[25]
Hemp	GVLY, IEE, LGV, RVR	ACE inhibitor	[26]
Rapeseed/Canola	-	antioxidant	[27]
Rapeseed/canola	FQW, FRW, CPF	ACE inhibitor	[28]
Rapeseed/canola	GHS, RALP, LY	ACE inhibitor	[29]
Rapeseed/canola	EFLELL	hypolipidemic	[30]
Sesame	SYPTECRMR	antioxidant	[31]
Soybean	-	antioxidant	[32,33]
Soybean	VLIVP	ACE inhibitor	[34]
Soybean	VHVV	ACE inhibitor	[35]
Soybean	-	antimicrobial	[36]
Soybean	PGTAVFK, IKAFKEATKVDKVVVLWTA	antimicrobial	[37]
Soybean	LPYP, IAVPTGVA, IAVPGEVA	hypocholesterolemic	[38]
Soybean	YVVNPDNDEN, YVVNPDNNEN	hypocholesterolemic	[39]
Soybean	-	immunomodulatory	[40]
Sunflower	YFVP, SGRDP, MVWGP, TGSYTEGWS	anti-inflammatory, immunomodulatory	[41]

## Data Availability

The data used to support the findings of this study can be made available by the corresponding author upon request.

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
