# Peer review of "Recent Trends in the Application of Oilseed-Derived Protein Hydrolysates as Functional Foods"

_foods, 2023, doi:10.3390/foods12203861_

Round 1

Reviewer 1 Report

Oilseed-derived proteins have emerged as an excellent alternative to animal sources for the production of bioactive peptides which possess  health-promoting and disease-preventing effects being used as ingredients for the development of functional products.

The current paper covers all this topics, but needs some improvement.

Oilseed crops are used in the food industry, e.g. for the production of butter, margarine- line 54, Butter is only made from animal fat.

Figure 1. Beneficial effects of Bioactive Peptides could include a schematic representation of the compounds structure in the middle, instead.

In Lines 88-105, 173 references should be included.

divide section 2. Biological Activity of Oilseed-Derived Bioactive Peptides, in sub sections.

The review could/should also revisit the isolation methods of bioactive peptides from the meals. information from lines 288-290 could be also included here.

in figure 2 ,,beverages,, is incorrect.

Chapter 3. Use of oilseed-derived protein hydrolysates as functional foods, needs to  be divided in subsections based on different food products, as in the figure.

Author Response

Thank you very much for taking the time to review this manuscript. Please see the attachment.

Reviewer 2 Report

The review was well-designed and contained enough information. Its section is directly related to its topic and its references too. However, it is suggested that the abstract be revised and enlarged according to the rules.

Author Response

(The authors gave the same response as above.)

Reviewer 3 Report

The manuscript does not sound to be a review article due to lack of vast research results and hence needs major revision.

English should be more concise across the manuscript.

Author Response

(The authors gave the same response as above.)

Reviewer 4 Report

Information stated as factual, particularly in a review paper, must have supporting evidence of scientific references or additional explanations.

Line(s)      Comment

14-23        Some specific examples of the health-promoting, disease-preventing, and/or contributions to functional foods should be included.

27-30        Reference(s) needed to substantiate the information in these sentences.

38-40        Reference(s) needed to substantiate the information in these sentences.

51-53        Reference(s) needed to substantiate the information in these sentences.

55-58        This sentence is not accurate as soybeans, cottonseeds, and peanuts are more commonly grown than evening primrose and sesame.

84-95        Reference(s) needed to substantiate the information in these sentences.

96-97        Abbreviations and acronyms should be defined at first use.

164-173   Reference(s) needed to substantiate the information in these sentences.

194-198   Reference(s) needed to substantiate the information in these sentences.

223           No need to capitalize nitric oxide.

233           “discussed in the previous section, they”

231-239   Reference(s) needed to substantiate the information in these sentences.

244-247   Reference(s) needed to substantiate the information in these sentences.

255-256   Reference(s) needed to substantiate the information in this sentence.

345-346   Reference(s) needed to substantiate the information in this sentence as it was not previously established that there is increased consumer awareness of a healthy diet.

376-517   There are inconsistencies in the format of journal titles, i.e. capitalization, among the references.

Author Response

(The authors gave the same response as above.)
